# Wild-Type and SOD1-G93A SH-SY5Y under Oxidative Stress: EVs Characterization and Topographical Distribution of Budding Vesicles

Carolina Sbarigia [1,†], Simone Dinarelli [2,†], Francesco Mura [3], Luca Buccini [3,4], Francesco Vari [5], Daniele Passeri [3,4], Marco Rossi [3,4], Stefano Tacconi [1,6] and Luciana Dini [1,3,*]

1   Department of Biology and Biotechnology "C. Darwin", University of Rome Sapienza, 00185 Rome, Italy
2   Institute for the Structure of Matter (ISM), National Research Council (CNR), 00133 Rome, Italy
3   Research Center on Nanotechnology Applied for Engineering of Sapienza (CNIS),
    University of Rome Sapienza, 00185 Rome, Italy
4   Department of Basic and Applied Sciences for Engineering, University of Rome Sapienza, 00161 Rome, Italy
5   Department of Physiology and Pharmacology "V. Erspamer", University of Rome Sapienza, 00185 Rome, Italy
6   CarMeN Laboratory (INSERM 1060/INRAE 1397), Lyon-Sud Faculty of Medicine, LYON 1 University,
    69310 Pierre Bénite, France
*   Correspondence: luciana.dini@uniroma1.it
†   These authors contributed equally to this work.

**Abstract:** Extracellular vesicles (EVs) are important mediators of intercellular communication in several physiopathological conditions. Oxidative stress alters EVs release and cargo composition depending on the cell type and stimulus. Recently, most of the EVs studies have focused on the characterization of their cargo, rather than on the morphological features (i.e., size distribution, shape, and localization on the cell surface). Due to their high heterogeneity, to fully characterize EVs both the functional and morphological characterization are required. Atomic force microscopy (AFM), introduced for cell morphological studies at the nanoscale, represents a promising method to characterize in detail EVs morphology, dynamics along the cell surface, and its variations reflecting the cell physiological status. In the present study, untreated or $H_2O_2$-treated wild-type and SOD1-G93A SH-SY5Y cells have been compared performing a transmission electron microscopy (TEM) and AFM morpho-quantitative analysis of budding and released vesicles. Intriguingly, our analysis revealed a differential EVs profiling, with an opposite behavior and implying different cell areas between WT and SOD1-G93A cells, on both physiological conditions and after $H_2O_2$ exposure. Our results empower the relationship between the morphological features and functional role, further proving the efficacy of EM/AFM in giving an overview of the cell physiology related to EVs trafficking.

**Keywords:** extracellular vesicles; oxidative stress; atomic force microscopy; transmission electron microscopy; surface budding; topographical mapping



## 1. Introduction

Extracellular vesicles (EVs) are a heterogeneous group of membranous nano- and micro-particles released by several cell types, diffusing in body fluids and carrying several molecular cargos (i.e., proteins, lipids, and nucleic acids) to other cell targets [1]. At the beginning, it was believed that the main EVs function was removing unwanted material, but nowadays EVs are widely recognized as active players in intercellular communication in both physiological and pathological conditions [2]. Due to their high degree of heterogeneity, EVs are difficult to classify in distinct and uniform categories and the most studied are the following two main groups: exosomes, which are also recognized as small vesicles (40–160 nm in size) originating from the endocytic pathway, and microvesicles (MVs), or large vesicles (150 nm–1 μm in size), which are directly released from the plasma membrane budding. To study EVs, according to the "Minimal Information for Studies of

EVs" drawn up by the International Society of Extracellular Vesicles (ISEV) [3], several steps are required after their isolation, including morphological, quantitative, and molecular characterization. To the best of our knowledge, the current studies in the field of EVs are mainly focused on the molecular profile of EVs to understand their function, leaving the morphological and quantitative characterization in the background. Nevertheless, since morphological features may be strictly related to the functional role, an in-depth qualitative and quantitative characterization of the EVs morphological features may offer the first overview of their potential role related to the cell physiological status [4]. The EVs release may be affected both in terms of quantity and cargo composition during pathological alterations, including oxidative stress. Generally, reactive oxygen and nitrogen species (ROS and RNS) are produced at low concentrations to modulate several biological processes, such as gene expression, cytoskeleton remodeling, cell proliferation, migration, differentiation, and apoptosis, but also EVs trafficking [5]. Oxidative stress occurs when ROS and RNS are overproduced at the expense of the antioxidant defenses, damaging all the cell macromolecules (i.e., nucleic acids, carbohydrates, proteins, and lipids) and thus causing harmful effects on both cell structure and function [6]. Among the modifications induced on the cellular components by oxidative stress, lipid peroxidation drives changes in phospholipids dynamics with consequent membrane rearrangements and alterations on its fluidity and permeability [7]. Given this evidence, it is not surprising that, under oxidative stress, EVs trafficking results are altered depending on the cell type and stimulus, generally toward an increase in EVs release [8,9]. The overbalance of ROS may affect not only the EVs amount, but also their cargo, which can be enriched in oxidant or antioxidant molecules, spreading detrimental or protective signals to other cell targets [10]. For this reason, the knowledge of EVs molecular cargo is gaining growing importance to track the progression of several oxidative stress-associated diseases, including cancer, neurodegenerative, and inflammatory disorders [11]. However, next to the morphological characterization of released EVs that may change their size and shape under different physiopathological conditions, a still neglected aspect of EVs trafficking studies is the definition of the morphological changes of the cell surface dynamics occurring after a specific stimulus. High-resolution atomic force microscopy (AFM), recently emerging for the studies of EVs at the nanoscale, enables the performance of a multiparametric analysis and helps gain information concerning the morphological, topographical, biophysical, and biomechanical features of biological specimens [12–14]. In a previous piece of work, we demonstrated that atomic force microscopy (AFM) can perform a high-resolution analysis of both isolated EVs and whole cells, defining the plasma membrane budding profile that may be differentially altered on specific regions of the cell after $H_2O_2$ treatment [4]. Furthermore, AFM allows us to obtain not only the topographical distribution of budding vesicles, giving localized information that can be useful to define the cell physiological status, but also to quantify the budding entity exploiting a morphometric parameter, namely, the roughness. Exploiting a multiparametric approach availing two different microscopy techniques, such as the electron microscopy (EM) and AFM, it is possible to obtain multiple pieces of information on both isolated EVs fractions and vesicle budding directly from the cell plasma membrane and to define EVs trafficking dynamics under different physiological conditions. While in our previous work we investigated the behavior of wild-type (WT) SH-SY5Y cells [4], in the current study, we extended our analysis combining TEM and AFM techniques to perform a multiparametric characterization by comparing WT and transfected human SOD1-G93A SH-SY5Y cells, which are widely used to study oxidative stress and neurodegenerative disorders [15,16]. The main aim of this combined approach was to trace a profile and detect the changes of either the released EVs and the vesicles directly budding from the cell surface of both the WT and mutated cells, either untreated or treated with $H_2O_2$. The G93A single substitution in the Cu-Zn Superoxide Dismutase (SOD1), which is associated with neurodegenerative disorders and oxidative stress, causes a conformational disorder in the enzyme structure, with a tendency to form oligomeric structures that are responsible of a toxic gain of function related to protein misfolding [17]. Notwithstanding, the exact role of its toxicity is not fully

understood, and the influence of the mutation on EVs trafficking has not been yet explored. Intriguingly, our analysis highlighted a totally opposite behavior from the two cell types, both in basal conditions and after inducing oxidative stress, suggesting that different cells may display distinct responses in physiological conditions and under exposure to the same stimulus. Although the biological significance of these changes is still yet to be clarified, our results further confirm the validity of using a multiple morphometric approach to obtain important details about EVs trafficking in relationship with the cell physiological state, valorizing the morphological analysis next to the functional one.

## 2. Materials and Methods

### 2.1. Cell Culture and Treatments

Human neuroblastoma SH-SY5Y cells were provided by Carrì's group [18]. Cells were cultured in 4.5 g/L glucose DMEM (Sigma-Aldrich, St. Louis, MO, USA) with the following supplements: 10% heat-inactivated Fetal Bovine Serum (FBS) (Sigma-Aldrich, St. Louis, MO, USA), 2 mM L-glutamine (Corning, Manassas, VA, USA), 100 UI/mL penicillin/streptomycin (Corning, Manassas, VA, USA), and 10,000 U/mL of amphotericin B (Sigma-Aldrich, St. Louis, MO, USA). Cells were maintained at 37 °C in a humidified atmosphere (5% $CO_2$). Cells were maintained in 75 cm$^2$ flasks at an 80% confluence by passage every two days. $H_2O_2$ (Carlo Erba, Milan, Italy) at a final concentration of 100 μM was used to induce oxidative stress. All the treatments were performed by 1 h exposure to $H_2O_2$, with a subsequent recovery of 24 h in EVs-depleted medium. Then, cells were collected to perform all the analyses.

### 2.2. MTT Assay

MTT assay was performed following the protocol presented in [19]. After treatments, WT and SOD1-G93A SH-SY5Y cells were incubated for 3 h at 37 °C and 5% $CO_2$ with the MTT working solution (1 mg/mL) diluted in DMEM without supplements. After incubation, cells were rinsed with phosphate-buffered saline (PBS) and the formazan crystals were dissolved with dimethyl sulfoxide (DMSO) (Carlo Erba, Milan, Italy). The optical density (OD) was detected at the Multiskan™ FC Microplate spectrophotometer (Thermo Fisher Scientific, Waltham, MA, USA) at 570 nm wavelength.

### 2.3. EVs Isolation by Ultrafiltration Combined with Size-Exclusion Chromatography (UF/SEC)

For UF/SEC, 20 mL of medium was collected in Amicon Ultra-15 Centrifugal Filter Units (MWCO = 10 kDa; Merck Millipore, Billerica, MA, USA) and concentrated up to 500 μL by sequential centrifugation at 4000× *g*. The obtained concentrate medium was collected, diluted with PBS if necessary, and fractionated by SEC, as described by Böing et al. [20]. Briefly, up to 1 mL concentrated medium was placed in a 10 mL Sepharose CL-2B column (GE healthcare, Little Chalfont, UK), using PBS as eluent. The EVs enriched fractions were then collected and further concentrated to a volume of 100 μL on an Amicon Ultra-4 Centrifugal Filter Unit (MWCO = 10 kDa; Merck Millipore) by several steps of centrifugation at 4000× *g*. The newly concentrated EVs fractions were used for further analysis.

### 2.4. Transmission Electron Microscopy

Sample preparation for transmission electron microscopy (TEM) analysis consisted in fixation of the cells with 2.5% glutaraldehyde in 0.1 mol/L cacodylate buffer pH 7.4 (one hour at ice temperature) and eventually post-fixed with 1% $OsO_4$ in the same buffer (two hours at ice temperature). Cell fixation was followed by dehydration with ethanol (25%, 50%, 70%, 90%, and 100%). Finally, cells were embedded in Spurr resin. TEM specimens were 60 nm thick sections. TEM setup was a Zeiss Auriga (Zeiss, Oberkochen, Germany) equipped with a STEM module, operating at 20 kV. Isolated EVs from all experimental groups were fixed with 0.1% paraformaldehyde (PFA) in DPBS (30 min at

room temperature), stained with 2% uranyl acetate (7 min at room temperature), loaded on 200 mesh grids, and finally analyzed through TEM operating at 20 kV.

### 2.5. Western Blot Analysis

To obtain the proteins, cells were lysed in ice for 1 h in RIPA buffer (NaCl 150 mM, Tris-HCl 50 mM pH 8, MgCl$_2$ 2 mM, SDS 0.1%, Deoxycholic Acid 0.5%, and NP40 1%) completed with phenyl-methyl sulfonyl fluoride (PMSF) and protease inhibitor cocktail (Thermo Fisher Scientific, Waltham, MA, USA). After incubation, cell lysates were then sonicated and pelleted by centrifugation for 10 min at 13,000× $g$ at 4 °C. Supernatants were collected in new tubes, and the protein dosage was performed using BCA assay kit (Thermo Fisher Scientific, Waltham, MA, USA). A total of 20 µg of proteins of EVs and cells were used for Western blotting analysis. Proteins were separated by SDS-PAGE (10–12% polyacrylamide, SureCast$^{TM}$ Acrylamide Solution, Thermo Fisher Scientific, Waltham, MA, USA) and transferred to nitrocellulose membranes. Membrane blocking was performed for 1 h at room temperature in 3% bovine serum albumin (BSA) diluted in Tris-buffered saline containing 0.1% Tween 20 (TBST). The membranes were incubated overnight at 4 °C with CD63 (1:1000 dilution, Santa Cruz Biotechnology, Dallas, TX, USA), Annexin1 (1:1000 dilution, Santa Cruz Biotechnology, Dallas, TX, USA), and Calnexin (1:1000 dilution, Abcam, Milan, Italy) primary antibodies diluted in 3% BSA. After several washes in TBST, membranes were incubated with a goat anti-mouse IgG (dilution 1:5000; Bethyl Laboratories, Montgomery, TX, USA) secondary antibody, diluted in TBST in 3% BSA for 1 h at room temperature. The detection of the immunoreactive bands was performed using commercial enhanced chemiluminescence reagent (ECL) (WesternBright$^{TM}$ Quantum, Advansta Inc., San Jose, CA, USA). Finally, the density of the detected bands was measured by densitometric analysis (ImageLab Software, version 6.1).

### 2.6. Atomic Force Microscopy Preparation Procedure, Imaging, and Analysis

In order to perform atomic force microscopy (AFM) analysis, untreated and H$_2$O$_2$-treated by human neuroblastoma SH-SY5Y cells were previously seeded in 6-well multiwells with coverslips at a concentration of $8 \times 10^4$ cells/cm$^2$, and incubated overnight in a 5% CO$_2$ humidified atmosphere, at 37 °C. After washing cells twice in DPBS and fixed in 1% glutaraldehyde in 0.1 mol/L cacodylate buffer pH 7.4 (30 min at ice temperature). Then, two additional and subsequent washing steps in DPBS were performed, followed by two final washing steps in bi-distilled water. As for the preparation of the isolated microvesicles-enriched fractions, they were fixed with 0.1% paraformaldehyde (PFA) in DPBS (30 min at room temperature) and then diluted in water [21]. AFM characterization was performed by acquiring morphological images of the samples in contact mode using a Multimode AFM setup (Bruker Inc., Billerica, MA, USA) that was equipped with standard triangular AFM cantilevers (DNP Bruker Inc., nominal spring constant 0.06 N/m and nominal tip radius 10 nm). The static applied load during the scanning was less than 1 nN in order to minimize the possibility of sample damaging, tip wear, and contamination. In order to obtain comparable images of the different samples, the scan area was first 60 µm × 60 µm when imaging cells, and then subsequent scans on areas of 20 µm × 20 µm were performed to magnify regions of interest. Representative 3D reconstructions and mapping of the cell areas (3 × 3 µm$^2$) were performed with an image analysis software (Gwyddion, version 2.62). Quantification of cell surface roughness (Rq) was performed using ImageJ Software. For statistical purposes, on the mapped regions (nuclear and perinuclear to cellular edges), squared 3 × 3 µm$^2$ areas were randomly selected, and five measurements were taken from each sample.

*2.7. Statistical Analysis*

Results are expressed as means $\pm$ SD. Multiple comparisons were performed by two-way ANOVA. Comparisons between two groups were performed using a student's *t*-test. Differences between groups were considered statistically significant for $p < 0.05$. Comparisons between two distributions were analyzed using a Kolmogorov–Smirnov test (GraphPad Prism 9 software, GraphPad Software, San Diego, CA, USA).

## 3. Results

*3.1. TEM Analysis Reveals That Oxidative Stress Differently Affects EVs Secretion of WT and SOD1-G93A SH-SY5Y Cells*

To induce mild oxidative stress conditions, both WT and SOD1-G93A SH-SY5Y cells were treated with 100 µM of $H_2O_2$ for 1 h. The lower concentration was selected to induce a stimulus sufficient to induce damage without killing cells and, at the same time, allowing EVs release (Figure S1). To check if EVs release effectively occurred, TEM was performed to analyze whole cells and EVs isolated fractions (Figures 1 and 2). The whole cell analysis revealed the presence of multivesicular bodies and plasma membrane budding on both WT and SOD1-G93A cells (Figure 1). Intriguingly, while in WT cells the budding seems clearly enhanced after $H_2O_2$ treatment (Figure 1A) with respect to the untreated counterpart, SOD1-G93A cells display a prominent budding even in physiological conditions (Figure 1B). These results represent the first evidence of a differential EVs release between the WT and SOD1-G93A cells, which is further confirmed by the isolated EVs analysis. After the isolation by ultrafiltration combined with size-exclusion chromatography (UF/SEC), the TEM analysis was extended to EVs isolated fractions from all the experimental groups (Figure 2). TEM micrographs of EVs obtained from both WT and SOD1-G93A cells revealed the presence of vesicles showing the EVs typical round-shaped morphology (Figure 2A). Moreover, the size distributions of these EVs indicated that, while EVs derived from untreated WT SH-SY5Y cells display a narrow range (20–200 nm) with a peak of 40 nm, after treatment, there is a shift toward bigger sizes (50–300 nm, with a peak of 100 nm), thus determining a significant increase in the average diameter with respect to the untreated (Figure 2B). On the contrary, EVs isolated from untreated SOD1-G93A SH-SY5Y already display a wide size distribution (from 40 to 280 nm, with 150 nm peak), while after $H_2O_2$ treatment, there is a sharp reduction in the size range (from 40 to 160 nm, with 80 nm peak) and consequently, of the average diameter (Figure 2B). Finally, a Western blot analysis of EVs specific markers, i.e., Alix and CD63, confirmed the presence of EVs in the isolated fractions of all the experimental groups, which is further supported by the expression of the endoplasmic reticulum integral protein Calnexin only in cell lysates but not in EVs fractions (Figure 2C). Intriguingly, these results suggest that, while in WT cells the $H_2O_2$ treatment induces a massive production of EVs, especially larger EVs, SOD1-G93A cells display a prominent EVs release starting from basal conditions without any treatment. The differential profile of released EVs with no treatment suggests a different behavior between WT and mutated cells not only in response to a given stimulus, but also in basal conditions.

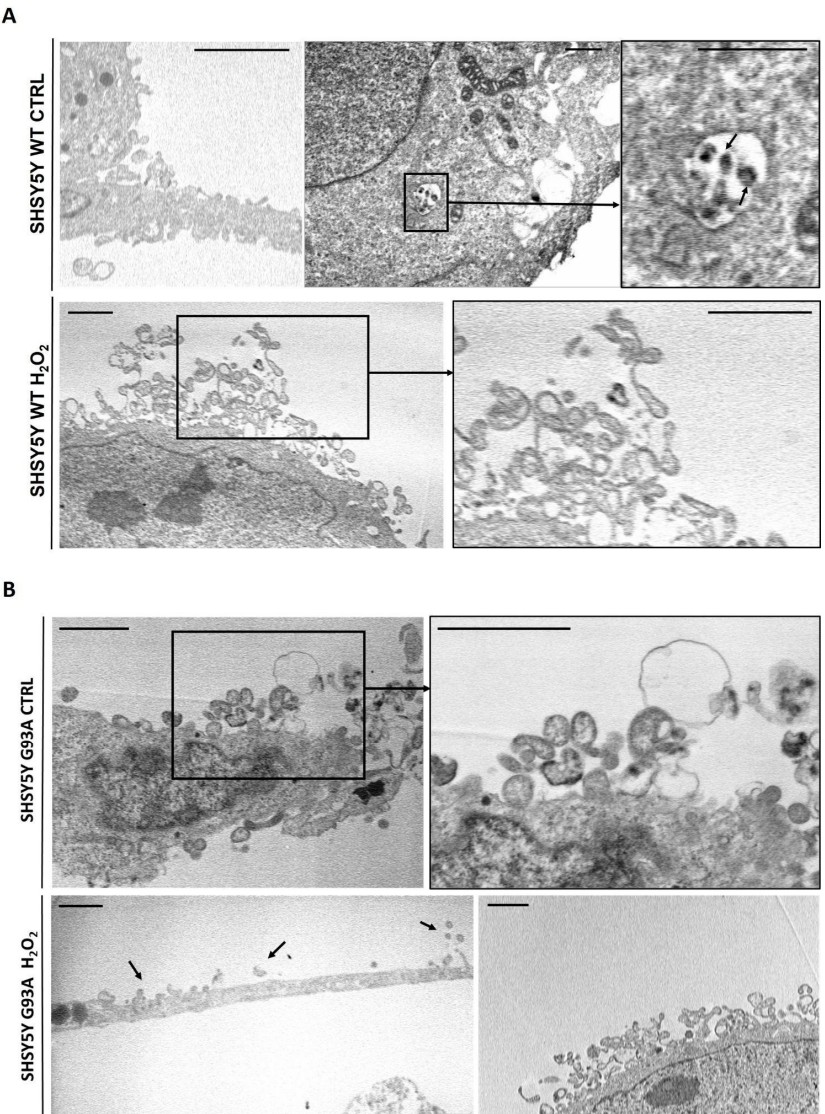

**Figure 1.** TEM micrographs of (**A**) WT SH-SY5Y cells untreated (control, **upper** panel) or treated with 100 μM of $H_2O_2$ (**B**) SOD1-G93A SH-SY5Y cells untreated (**upper** panel) or treated with 100 μM of $H_2O_2$. Scale bars = 2 μm. Arrows indicate the details of budding vesicles.

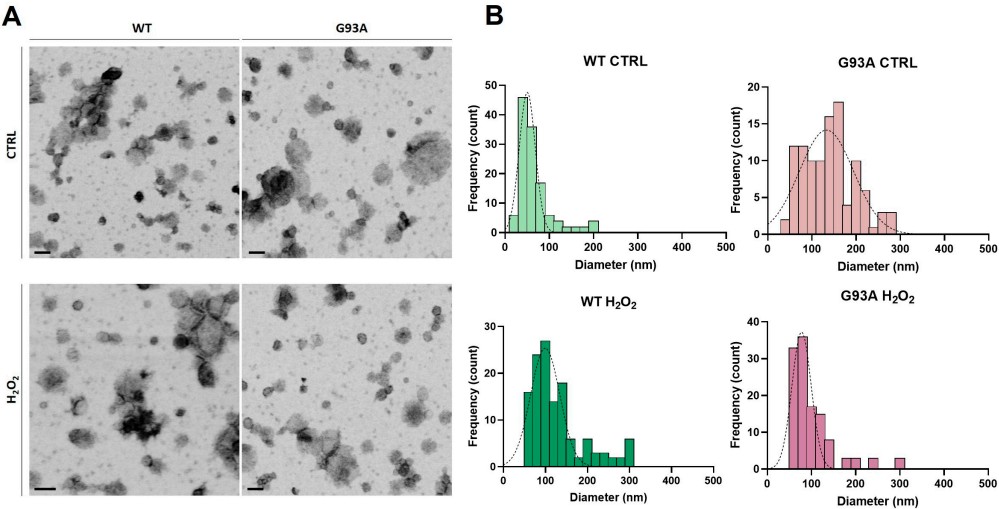

**Figure 2.** *Cont.*

C

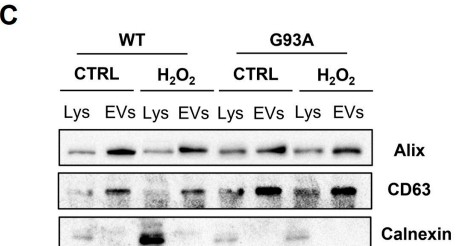
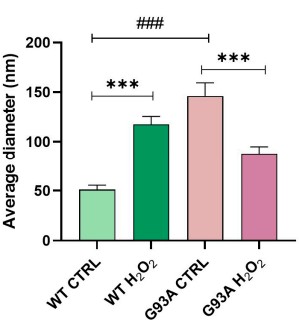

**Figure 2.** Morphological and quantitative characterization of extracellular vesicles released from untreated and $H_2O_2$-treated WT and SOD1-G93A SH-SY5Y cells. (**A**) TEM analysis of EVs released from SH-SY5Y WT and SOD1-G93A cells, untreated (control) or treated with $H_2O_2$ 100 μM. Scale bars = 100 nm. (**B**) Size distribution and average diameter of the EVs from all experimental groups, Ntot = 59, 47, 43, and 47. Values represent the mean ± SD; statistical analysis was carried out using one-way ANOVA with Tukey's multiple comparisons test. *** $p < 0.001$, when compared with the control group; ### $p < 0.001$, when compared between WT and G93A groups. Lys = cell lysates. (**C**) Alix, CD63, and Calnexin protein expression from SH-SY5Y WT and SOD1-G93A cells, untreated (control) or treated with $H_2O_2$ 100 μM, performed by Western blot.

*3.2. AFM Topographical Analysis Shows a Differential Profiling of Budding Vesicles along the Surface of WT and SOD1-G93A SH-SY5Y Cells under Oxidative Stress*

In a previous work, we validated the use of AFM to analyze the topographical distribution of vesicles directly budding from the plasma membrane, demonstrating that AFM allows us to acquire more consistent data and appreciate details that can be mislaid due to the harsh preparation method required for the transmission and scanning electron microscopy (TEM and SEM) [4]. After performing an AFM topography of all the experimental groups, it emerged that both WT and SOD1-G93A SH-SY5Y cells displayed a massive budding along the plasma membrane (Figure 3). While in WT cells vesicle budding appears predominantly on the cell bodies (Figure 3A), SOD1-G93A cells display a very irregular surface with a prominent blebbing also along the protrusions (Figure 3B). Intriguingly, the presence of vesicles released by the SOD1-G93A protrusions has also been detected by TEM analysis (Figure 1B), thus indicating a certain degree of comparability between the two techniques despite the different resolutions. After treatment with $H_2O_2$, both WT and SOD1-G93A cells apparently show an increased budding on the cell surface (Figure 3A,B), but again, with differences on the size distributions. In basal conditions (untreated group), buds from WT cells display a size distribution range between 200 and 500 nm, which is in agreement with the large vesicles dimensions, with a shift toward a broader distribution from 200 to 600 nm and bigger sizes after $H_2O_2$ treatment (Figure 3C). Indeed, the median diameter of WT budding vesicles increases from 295.2 ± 3.4 nm for the untreated group to 308 ± 3.8 nm for the $H_2O_2$-treated group (Figure 3E). Interestingly, this tendency is reversed when dealing with SOD1-G93A budding vesicles, showing a size distribution between 200 and 400 nm that remains approximately unaltered after $H_2O_2$ treatment (Figure 3D). However, when comparing the median diameters, $H_2O_2$-treated SOD1-G93A cells produce mainly smaller vesicles (262.5 ± 3.8 nm), not only with respect to the untreated group (287.5 ± 7.7 nm), but also to the $H_2O_2$-treated WT counterpart (Figure 3E). Although AFM allows us to study the topography of the cell surface, thus mainly representing the large vesicles enriched fractions (i.e., vesicles directly budding from the plasma membrane) rather than the entire EVs population released by the cells, it displays a good agreement with the TEM analysis, thus corroborating its validity to study variations in EVs trafficking in response to a given stimulus. Indeed, AFM analysis further confirmed the opposite behavior and the differential EVs profile, in terms of size as well as the distribution along the cell surface, between WT and SOD1-G93A SH-SY5Y cells on both physiological (untreated) and altered (after $H_2O_2$ exposure) conditions.

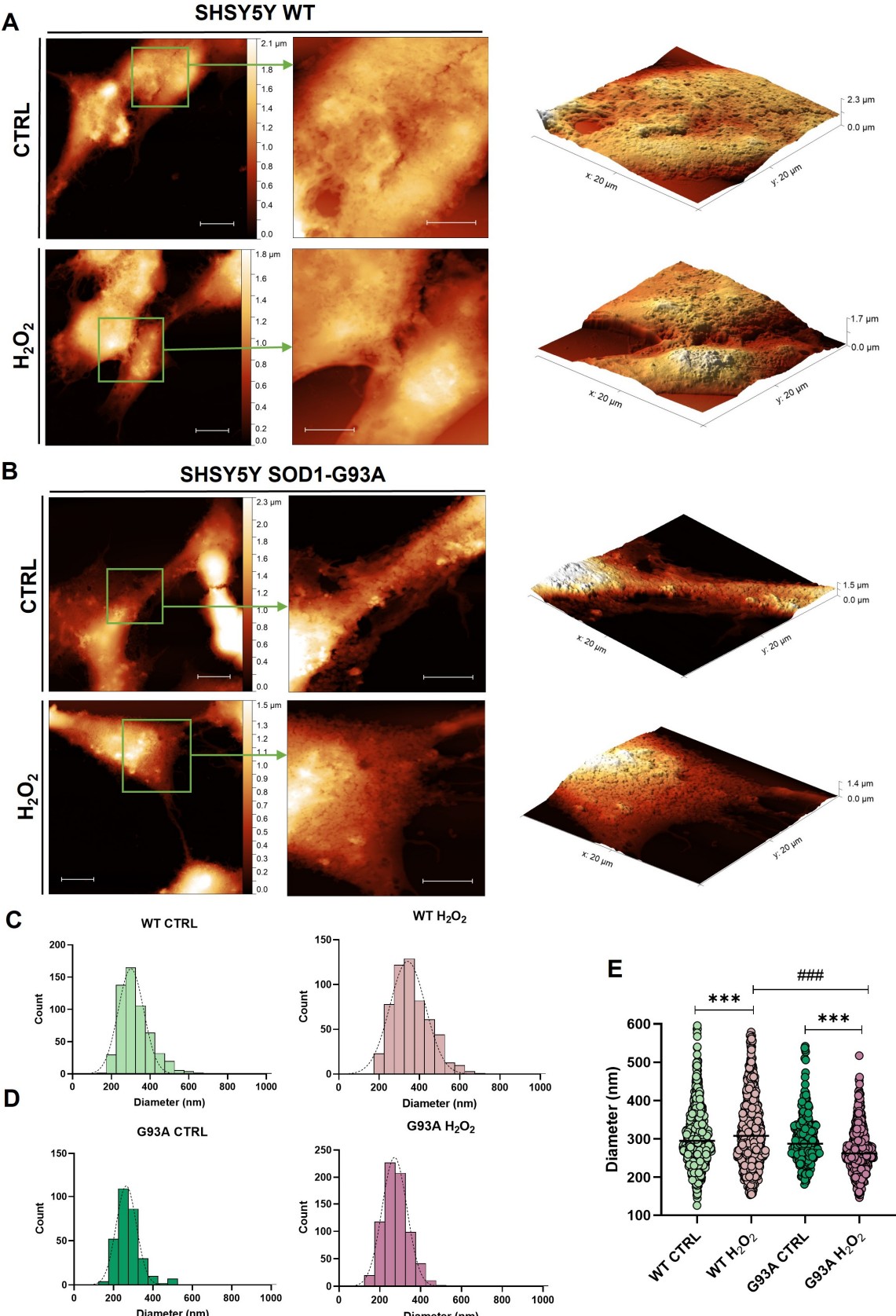

**Figure 3.** AFM topography, 3D reconstructions, and relative size distributions of SH-SY5Y cells. (**A**) SH-SY5Y WT untreated (CTRL) and treated with $H_2O_2$ 100 μM; (**B**) SH-SY5Y SOD1-G93A untreated (CTRL) and treated with $H_2O_2$ 100 μM. Scale bars = 10 μm for the whole cells and 4 μm for

the magnifications. (**C**) Size distributions of budding vesicles from WT SH-SY5Y cells, untreated (**left** panel, Ntot = 569) or treated with $H_2O_2$ (**right** panel, Ntot = 566). (**D**) Size distributions of budding vesicles from SOD1-G93A SH-SY5Y cells, untreated (**left** panel, Ntot = 258) or treated with $H_2O_2$ (**right** panel, Ntot = 470). (**E**) Direct comparison of the median diameters of budding vesicles from WT (295.2 ± 3.4 nm for CTRL and 308 ± 3.8 nm for $H_2O_2$) and SOD1-G93A (287.5 ± 7.7 nm for CTRL and 262.5 ± 3.8 nm for $H_2O_2$) SH-SY5Y cells. Values represent the mean ± SD; statistical analysis was carried out using one-way ANOVA with Tukey's multiple comparisons test. *** $p < 0.001$, when compared with the control group; ### $p < 0.001$, when compared between WT and G93A groups.

### 3.3. AFM Topographical Mapping of Central and Distal Regions of WT and SOD1-G93A SH-SY5Y Cells: Detecting the Details of a Differential Budding Profile

When cells are exposed to particular stimuli, such as oxidative stress, localized changes in vesicle budding may occur. This happens because EV release is a finely regulated process in which membrane rearrangements take place in specific regions of the cell in response to environmental conditions [11]. In our previous work, we demonstrated that a zonal morphometric analysis of budding vesicles performed by AFM provided different information than an analysis of the whole cell surface [4]. In the same manner, we selected confined areas ($3 \times 3$ μm$^2$), specifically on the nuclear regions (nuclei, Nu; blue dotted circles in Figures 4A and 5A) and the peripheral areas (perinuclear-to-cellular edge, Pn/CE; green dotted circles in Figures 4A and 5A) of the cells. Then, we first analyzed the size distribution of budding vesicles in the selected areas for both WT and SOD1-G93A SH-SY5Y cells. For the untreated group of WT cells, the size distributions range over 200–600 nm, with a peak of ~350 nm on both the nuclear and peripheral regions (Figure 4C,E, upper panels). The same diameter range can be observed for $H_2O_2$-treated WT cells (Figure 4C,E, lower panels). On the contrary, SOD1-G93A cells display more variability in size distributions according to their conditions, especially under oxidative stress induction. In the untreated group, the size distribution in the nuclear region ranges between 200 and 600 nm, with a ~350 nm peak (Figure 5C, upper panel), getting tighter (from 200 to 400 nm) after $H_2O_2$ treatment (Figure 5C, lower panel). Notwithstanding, the peripheral regions display a distinct behavior, with a size distribution ranging from 150 to 450 nm (with a ~300 nm peak) in basal conditions (Figure 5E, upper panel) and becoming broader after the induction of oxidative stress (from 200 to 600 nm, with a ~350 nm peak) Figure 5E, lower panel. Furthermore, while the median diameter values display a significant increase in the peripheral regions of WT budding vesicles (from 297.4 ± 4.8 nm to 334 ± 5.3 nm) after $H_2O_2$ treatment, there is instead a sharp reduction (from 343.8 ± 6.1 nm to 275.4 ± 4.9 nm) localized in the nuclear regions of SOD1-G93A cells (Figure 6A). Interestingly, this reduction in the median diameter of budding vesicles in the nuclei of $H_2O_2$-treated SOD1-G93A cells is also significant with respect to the WT treated counterpart (Figure 6A). These results indicate that, under oxidative stress conditions, while in WT SH-SY5Y cells the vesicle budding is preferentially affected in the distal regions of the cell, in SOD1-G93A cells the variation is primarily charged to the nucleus, further suggesting that mutated cells display a differential response to $H_2O_2$ exposure in comparison to the WT cells.

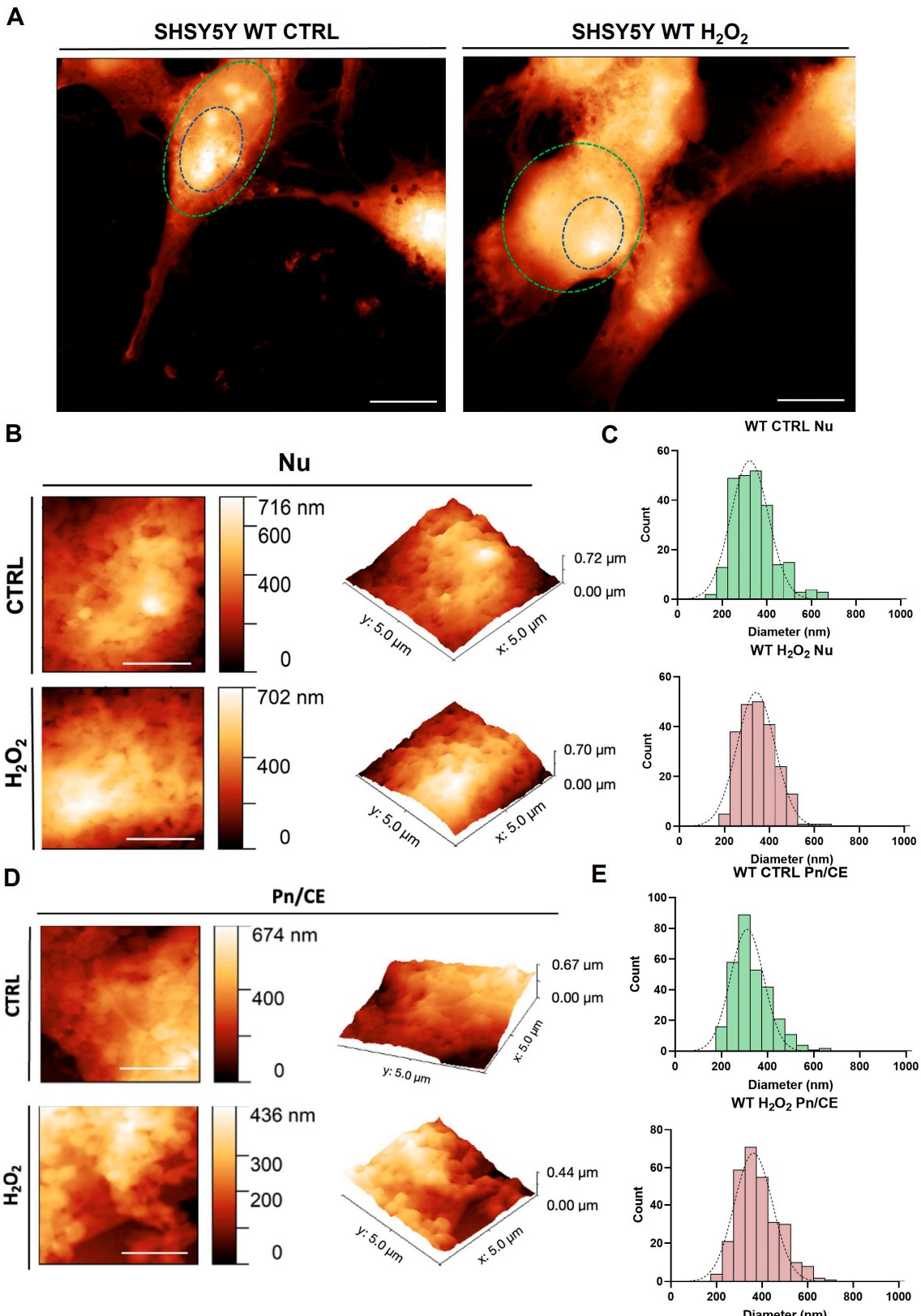

**Figure 4.** Mapping of the surface of CTRL (**A**) and $H_2O_2$ (**B**) WT SH-SY5Y cells in the nuclear zone (**B**) and perinuclear-to-cellular edge regions (**D**), with relative 3D reconstructions and size distributions (**C**,**E**), Ntot = 243, 273, 298, and 292. Scale bars = 10 μm for whole cells micrographs; 1 μm for magnifications of the 3 × 3 μm² mapped regions. Nu = nuclear zone (blue circles); Pn/CE = perinuclear-to-cellular-edge regions (green circles).

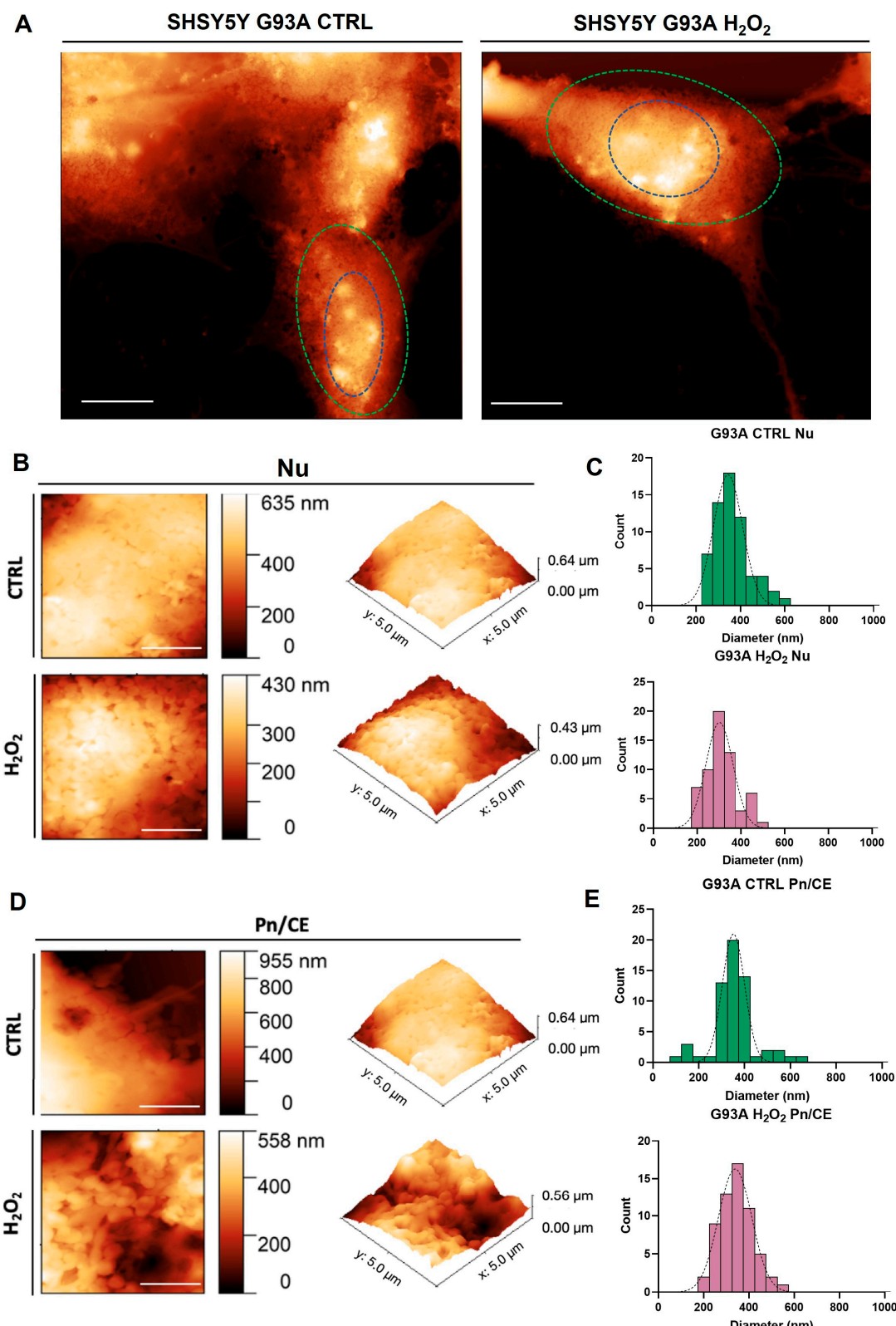

**Figure 5.** Mapping of the surface of CTRL and $H_2O_2$ SOD1-G93A SH-SY5Y cells (**A**) in the nuclear zone (**B**) and perinuclear-to-cellular edge regions (**D**), with relative 3D reconstructions and size distributions (**C**,**E**), Ntot = 62, 60, 60, and 60. Scale bars = 10 μm for whole cells micrographs; 1 μm for magnifications of the mapped regions. Nu = nuclear zone (blue circles); Pn/CE = perinuclear-to-cellular-edge regions (green circles).

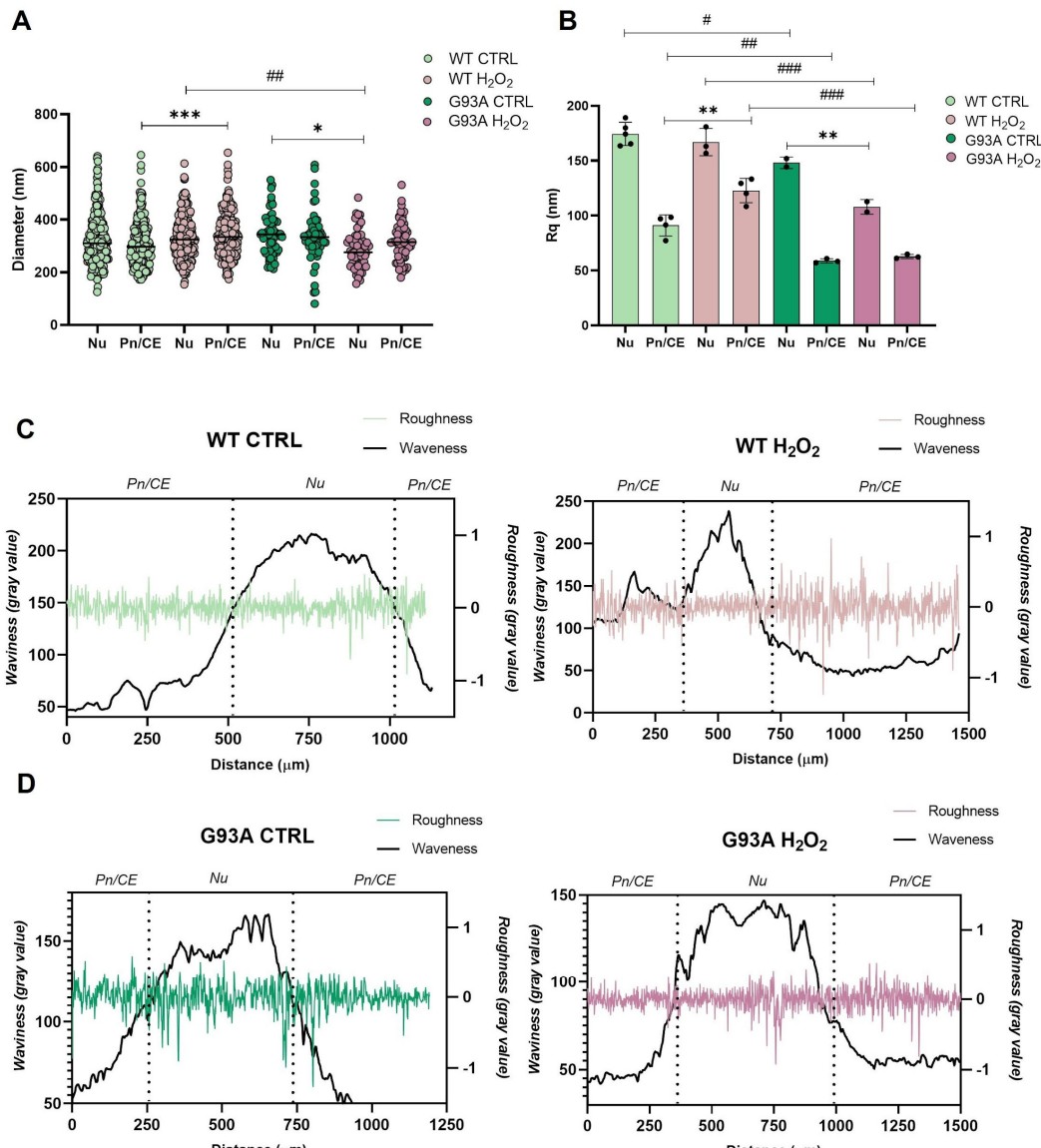

**Figure 6.** (**A**) Direct comparison of the median diameters of budding vesicles of mapped regions ($3 \times 3\ \mu m^2$) from CTRL (Nu = 310.0 nm; Pn/CE = 297.4) and $H_2O_2$-treated (Nu = 323.6 nm; Pn/CE = 334.0 nm) WT and CTRL (Nu = 343.8 nm; Pn/CE = 333.2 nm) and $H_2O_2$-treated (Nu = 275.4 nm; Pn/CE = 314.5 nm) SOD1-G93A SH-SY5Y performed by AFM. (**B**) Rq measurements of mapped regions of all the experimental groups. Values represent the mean $\pm$ SD; statistical analysis was carried out using one-way ANOVA with Tukey's multiple comparisons test. * $p < 0.05$, ** $p < 0.01$ and *** $p < 0.001$, when compared with the control group; # $p < 0.05$, ## $p < 0.01$ and ### $p < 0.001$, when compared between WT and G93A groups. (**C,D**) Roughness and waviness (gray values) measurements performed on all the experimental groups (**upper** panels for WT cells, lower panels for SOD1-G93A cells). Vertical dotted lines separate nuclear from peripheral regions. Nu = nuclear zone; Pn/CE = perinuclear-to-cellular edge regions.

### 3.4. Quantifying the WT and SOD1-G93A SH-SY5Y Cells Differential Budding Profile: Analysis of the Surface Roughness

High-resolution AFM analysis allowed the detection of localized variations in the budding profile of different cell lines, such as WT and mutated in SOD1-G93A SH-SY5Y, after a given stimulus (i.e., $H_2O_2$). However, the advantage of performing an AFM analysis consists not only in determining the topography of the sample at the nanoscale, but also in obtaining multiple pieces of information about its biophysical properties. We extended our

analysis to the previously mapped regions (Nu and Pn/CE) with a quantitative measurement of the budding entity by using the roughness (Rq), which was strictly correlated with the morphological variations of the plasma membrane, as demonstrated before in our previous work [4,22]. We depicted multiple confined areas ($3 \times 3 \ \mu m^2$) in the nuclear and in the peripheral regions to obtain a representative average value of the roughness. The obtained values are represented in Figure 6B. Interestingly, the average Rq values follow the same trend as for the variations previously found in the size distributions. In WT cells, there is a significant increase in the average Rq value in the Pn/CE between untreated (91.09 nm) and $H_2O_2$-treated (122.8 nm) (Figure 6B). Accordingly, in the roughness/waviness charts, which are represented in the cell profiles on the lateral scale (black lines) and the local roughness (colored frequencies), WT untreated cells display a similar roughness profile (light green frequencies, Figure 6C, left panel) with respect to the $H_2O_2$-treated (light pink frequencies, Figure 6C, right panel) in the nuclear zones. However, when looking at the perinuclear-to-cellular edge regions, there is a noticeable increase in the roughness gray values in WT $H_2O_2$-treated cells with respect to the untreated (Figure 6C). On the contrary, SOD1-G93A cells again show an opposite behavior, with a significant decrease in the average Rq values of the $H_2O_2$-treated nuclei (148.2 nm vs. 108 nm) in comparison to the untreated ones (Figure 6B). Correspondingly, the roughness/waviness profiles reveal a flattening of the roughness gray value in $H_2O_2$-treated SOD1-G93A cells (dark pink frequencies, Figure 6D, right panel) with respect to the untreated counterpart (dark green frequencies, Figure 6D, left panel). These results suggest that the variations observed in the size distributions find an agreement with the roughness values, thus indicating that the surface budding is affected not only concerning the vesicles size, but also in terms of the budding entity. Furthermore, when directly comparing the two cell lines (WT vs. SOD1-G93A), significant differences emerge in the Rq mean values, and consequently in the budding entity, either in the nuclear regions and the peripheral regions or under physiological and oxidative stress conditions (Figure 6B). This indicates that, in basal conditions, WT and SOD1-G93A cells display a different budding vesicles profile along the plasma membrane surface, which further undergoes opposite variations when exposed to $H_2O_2$, concerning different districts of the cell and an opposite trend.

## 4. Discussion

Extracellular vesicles (EVs) are heterogeneous, membrane-enveloped particles originating from disparate cell types, carrying several active biomolecules toward nearby or distant targets, which are consequently responsible for intercellular communication in distinct physiopathological conditions [1]. The evidence of EVs as the leading mediators of cell-to-cell crosstalk has directed the attention of the scientific community toward a better understanding of EVs role in several cell physiology alterations and disorders, including cancer, inflammatory, neurodegenerative, and metabolic diseases [23–25]. Most of these pathologies are associated with oxidative stress, an intricate physiological alteration emerging when an excess of pro-oxidant stimuli takes down the cell antioxidant defenses, leading to harmful signals and damage of the macromolecules mining the cell structure and integrity. In particular, oxidative stress alters either EVs release or content, generally toward an increase in EVs production and with effects ranging from protective to detrimental depending on the microenvironment (i.e., cell type and stimulus received) [5]. However, most of the studies focused primarily on the characterization of the EVs cargo to unravel their function, disregarding the morphological characteristics and the changes of the cell surface dynamics occurring under oxidative stress that may instead offer the first evidence of an altered physiological status of the cell. Therefore, a better description and definition of EVs structural properties and dynamics on the cell surface may offer precious information about their potential function and should not be underestimated. According to the ISEVs "Minimal Information for Studies of EVs" [3], to have a comprehensive view of EVs features, morphological and quantitative characterization are needed as well as the definition of their molecular cargo. In a previous work, we exploited a combined

morphological characterization approach, performing a SEM/AFM comparative analysis, to validate the AFM as a method to study EVs as well as the EM conventional one [4]. In the present work, we corroborate the unique capability of AFM to detect localized changes in EV dynamics under a given stimulus, providing accurate information about the cell's physiological status. In particular, by performing a morphological characterization by TEM of isolated EVs and AFM for budding vesicles, we highlighted that the SOD1-G93A mutation significantly alters the profile of both released and budding EVs of human neuroblastoma SH-SY5Y cell lines in comparison to the wild-type counterpart, both in physiological and oxidative stress conditions (induced with $H_2O_2$ treatment). Starting from the TEM analysis of both whole cells and released vesicles, it appears that, while in WT cells the exposure to $H_2O_2$ induces a massive EV release and a wider size distribution, in SOD1-G93A there is a distinct behavior, with the presence of a prominent budding of the plasma membrane already in basal conditions. Taken together, TEM results indicate not only a different responsiveness of the cells to $H_2O_2$ exposure, but also different morphological features and EVs profile in basal conditions. Oxidative stress is widely recognized to differently modulate EV trafficking, both in terms of quantity and cargo, depending on the cell type and stimulus given [11]. Despite this, TEM analysis clearly marks how morphological changes are quite evident between different cell types and how they occur starting from physiological conditions. The SOD1-G93A mutation is notably associated with oxidative stress [17,26,27]. However, to the best of our knowledge, the influence of this kind of mutation on EV trafficking has not yet been investigated. Interestingly, the presence of the SOD1-G93A mutation, in the absence of any external stimulus, seems to be sufficient to induce a differential budding in comparison to the wild-type cells, generating at the same time an opposite response to $H_2O_2$ treatment. EV release and cargo loading are connected and finely regulated processes, in which multiple biogenetic pathways act simultaneously, driving localized changes in lipid rafts and protein components that, in turn, modulate the membrane's curvature and fluidity [11]. Therefore, it is plausible that certain regions of the cells may be involved more than others when exposed to a specific stimulus. To further explore the differences in EV trafficking between WT and SOD1-G93A cells, we performed a high-resolution topographical analysis of budding phenomena by AFM, considering the variations in the blebbing dynamics charged to the whole cells and, specifically, the single contribution of localized regions (nuclear and perinuclear/cellular edge areas). Intriguingly, AFM analysis of budding vesicles on the entire cell surface confirmed the same trend as for the isolated EVs analyzed by TEM, with a significant increase in the median diameters from untreated to $H_2O_2$-treated WT cells, and a specular decrease from untreated to $H_2O_2$-treated SOD1-G93A cells. Moreover, in agreement with the TEM analysis, AFM micrographs revealed a high concentration of budding vesicles localized on both the cell bodies and protrusions of SOD1-G93A cells, unlike the wild-type cells in which the blebbing seems more confined in the cell bodies. These results suggest that the alterations on EVs budding profiles, on both physiological and stress conditions, may occur prior to their release in the external environment. Furthermore, it is plausible that the main contribution to the variation observed in the SOD1-G93A-released EV profile is given by the vesicles that are directly budding from the plasma membrane. During the membrane remodeling process occurring prior to EVs release, the molecular machinery involved in EVs biogenesis creates clusters resulting in ordered microdomains [28]. For this reason, to understand which cellular district was mostly affected by changes in the budding phenomena, we mapped the cells considering two main parts, the nuclear region and the one covering from the perinuclear zone to the cellular edges [4]. Intriguingly, WT and SOD1-G93A SH-SY5Y cells showed not only a differential profile of budding vesicles in terms of size, but also in terms of budding entity, implying different districts of the cell. By using the membrane roughness (Rq), a morphometric parameter strictly related to cell surface arrangements, we were able to detect alterations in the plasma membrane disposition, thus quantifying the budding phenomena [4,22]. From AFM analysis on the mapped regions emerged the fact that wild-type cells go through a significant increase in

the budding entity and size distribution, preferentially in the peripheral regions, while the nuclear area remains approximately unaltered. Conversely, SOD1-G93A cells undergo a reduction in both the budding entity and vesicle size, specifically localized on the nuclear part, leaving the distal regions unaffected. Finally, SOD1-G93A cells generally display a reduced budding intensity in comparison to the WT cells in all the selected areas (nuclear and peripheral regions) and in all the experimental conditions (untreated groups and after $H_2O_2$ exposure). These results further demonstrate the power of AFM to get detailed information about the cell morphology with reference to size, intensity, and position of the budding phenomena, highlighting the differences between wild-type and SOD1-G93A mutated cells. The detected variations are not just reflecting a differential EV profile in terms of morphology, dynamics, and budding entities but may also have an impact on cargo selection, implying different interactions with other targets and consequently distinct functions. Taken together, these results prove the efficacy of AFM related not only to the morphological characterization, confirming its agreement with a conventional method such as TEM, but also to its ability to give precious details concerning the biophysical and mechanical characteristics of the sample, making it possible to discriminate between distinct biological specimens and different environmental conditions. Since recently it has been claimed that there is a relationship between morphology and function, it is pivotal to not underestimate the contribution of an accurate definition of the EVs morphological features. What remains to be elucidated are the mechanisms underlying these variations, and thus what biological pathway may be mostly responsible for the defective EV release in the presence of the SOD1-G93A mutation.

**Supplementary Materials:** The following supporting information can be downloaded at: https://www.mdpi.com/article/10.3390/applnano4010004/s1, Figure S1: MTT assay performed on WT and SOD1-G93A SH-SY5Y treated with $H_2O_2$.

**Author Contributions:** Conceptualization, L.D. and S.T.; formal analysis, F.M. and S.D.; data analysis and curation, S.D., C.S., S.T., F.V. and L.B.; writing—original draft preparation, C.S.; writing—review and editing, C.S., D.P., M.R., S.T. and L.D. All authors have read and agreed to the published version of the manuscript.

**Funding:** The authors acknowledge the financial support provided by the Lazio Region through the "Open infrastructures for research" Call, which allowed us to carry out this research through the ATOM project for providing access to state-of-the-art instrumentation [POR-FESR 2014-2020: DE G06705, 25/06/2028, project code 173-2017-17395].

**Data Availability Statement:** The raw data supporting the conclusions of this article will be made available by the authors, without undue reservation.

**Conflicts of Interest:** The authors declare no conflict of interest.

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
