# Peer review of "Wild-Type and SOD1-G93A SH-SY5Y under Oxidative Stress: EVs Characterization and Topographical Distribution of Budding Vesicles"

_2673-3501, doi:10.3390/applnano4010004_

Round 1

Reviewer 1 Report

Would the authors highlight the novelty of this work compared to their previous work (reference no.4)?

Should they explain why they used the same oxidative stress testing as in their previous work?

In my point of view, comparing the results with that of other inducer would enrich the study.

Author Response

In the current literature, several works used H2O2 to induce and investigate oxidative stress on human neuroblastoma SH-SY5Y cells [Zhang, J., et al, 2021, doi:10.3389/fnagi.2021.754956; Zhou, Y., et al, 2022, doi: 10.31083/j.fbl2701008; Wan, T., et al, 2022, doi: 10.1155/2019/8239642; to cite just a few]. We chose H2O2 as oxidative stress inducer for our previous work, already cited in the manuscript with the reference [4]. This work has the main aim to extend the previous one [4], in which all the analyses were performed only in wild-type cells, by introducing in the analysis the cells carrying the mutation in SOD1-G93A. The final part of the introduction has been rewritten in a small part to make clearer the aim of our work. However, the leading purpose of this paper, as emphasized in the last part of the introduction and recovered in the discussion, was to empower the validity of the use of AFM in detecting details that can reveal precious information about the cell physiological state; in this way, the oxidative stress represents just an example of alteration in the cell normal functionality.

Reviewer 2 Report

In the present study, Sbarigia et al. asked whether extracellular vesicles released by WT and SOD1-G93A SH-SY5Y cells in the presence or absence of oxidative stress (H2O2 treatment) exhibit differences at the quantitative and morphological levels. They used a combination of TEM and high-resolution AFM to analyze both isolated EVs and the cell surface, a methodology they described in a previous study (Sabrigia et al., Front Cell Dev Biol 2022). The authors main conclusions are that WT and SOD1-G93A cells exhibit different EV profiles, that involve different cell areas, both under physiological and oxidative conditions.

A reliable methodological pipeline to analyze when, how and which EVs are released by cells under specific conditions is certainly of great interest. However, I had many issues with the paper and in my opinion the data is too often over-interpreted and does not support the conclusions drawn by the authors.

- lines 24-25: "To date, most of the EVs.....features". This is a rather gratuitous claim; also 'morphological features' should be more clearly defined.

- line 48: "EXOs" is not a standard acronym for exosomes and is not used in the rest of the paper; it should be removed.

- lines 52-55: "Recently, the majority....background": same comment as above; no evidence for this claim which depends a lot on the fields of study.

- lines 76-80: again, the 'morphology' of EVs needs to be clearly defined as it can mean many things, from size, vesicle-surface markers, membrane composition, etc.

- line 117: please specify how the medium was depleted from EV (any QC check?); if commercially available, please give the reference.

- lines 149-150: I was surprised by the low concentration of PFA (0.1%) used and the short time (30 min) allowed for EV fixation. I am not sure this is effective as the norm is usually 1-2% PFA and fixation for at least 1-2h at 4°C. The authors should elaborate on this (or give appropriate references for this method of fixation).

- lines 200-206: Statistical analysis. In all figures, the authors do not specify how many particles they counted and included in their statistical analysis. When including a great numbers of vesicles in the analysis, artifactual 'statistical significance' is easy to obtain, although its physiological meaning can be questionable (see my comments below). Also, while I understood that multiple cell areas were analyzed (technical replicate), it was not clear how many different cells (biological replicates) were analyzed (are the cells shown in the figures representative of the population?).

- Figure 1, lines 215-217: no 'massive budding' is seen in the WT cells w/o H2O2. Representative images of the cell bodies and protrusions should be shown for all cell types and conditions.

- Figure 2: The EM images of EVs are of poor quality, suggest that the EV preparations are not sufficiently pure (+ lots of vesicle aggregates) and cannot be used to accurately determine size distributions. Any claimed differences here cannot be determined using this type of images. The authors should consider alternate methods such as NTA to analyze their EV populations and numbers (does the very low CD9 signal in G93A cells + H202 equals less vesicles?)

- line 280: 'prolongations' is not an adequate word; maybe replace with 'protrusions'?

- Figures 3-5: I don't understand how the authors can distinguish vesicle-budding (and measure vesicles diameters) from cell-surface alterations,  membrane blebbing or other cell-surface processes (e.g. macropinocytosis) in their AFM analysis. The authors conclude that statistically significant differences in vesicle size distributions  exist between the cells, cell areas and growth conditions. However (see my comment above), the measured differences are very small (e.g. line 346: 297,4 nm vs 334 nm; line 403: 148,2 nm vs 108 nm) and their significance questionable. We do not know the intrinsic error of the measurements and again statistical analysis should be conducted with care when analyzing lots of particles. Also, it is not clear how the position of the nucleus and the pericentral areas were determined.

Author Response

  1. Lines 24-25: The limit of words required in the abstract made quite difficult to do other specifications. However, in the field of extracellular vesicles, there are minimal requirements for the studies of extracellular vesicles (already cited in the article: Witwer KW et al, J Extracell Vesicles, 2021) and it is widely known that the functional role is generally referred as the molecular composition of the cargo, while the morphological features are the size distribution, shape and localization along the cell surface. We took this information for granted; however, to be more clear, we made the requested changes.
  2. Lines 48: the “EXOs” acronym was deleted as requested.
  3. Lines 52-55: The statement has not to be intended as absolute, but of course is referring to the field of extracellular vesicles and the current literature to the best of our knowledge. Since extracellular vesicles are emerging as important intercellular communication in several physiological and pathological conditions (see references 1,2,5,11,16,17,23), most of the studies in this field are focused mainly on their potential functions (in terms of the characterization of their cargo, role in the pathogenesis of several diseases in terms of cell of origin and cell targets, as well as potential therapeutic applications) rather than on morpho-quantitative analyses (i.e., changes in the size distribution, shape, and localization on the cell surface in different environmental conditions). The take home message should be that, since the EVs biogenesis and trafficking are altered in response to the different cellular microenvironment, in our opinion an in-depth morpho-quantitative analysis may offer precious information about the cell physiology  as well as the functional characterization and should not be underestimated. However, as not to make the reader fall into misunderstanding, we specified that the statement comes from the best of our knowledge concerning the field of EVs.
  4. Lines 76-80: Here we reported the following statement: “However, next to the morphological characterization of released EVs that may change their size and shape under different physiopathological conditions, a still neglected aspect of EVs trafficking studies is the definition of the morphological changes of the cell surface dynamics occurring after a specific stimulus“. As underlined here, as morphological features we are taking in consideration the size distribution, the shape and the cell surface dynamics (intended as the budding localization, detected with a high-resolution microscopy technique, and intensity of the budding directly correlated with the surface roughness, see references 4 and 15). In the field of extracellular vesicles, EVs surface markers and the membrane composition are not included in the EVs morphological features, but rather in the molecular characterization (see reference 3).
  5. (see the figure in the attached file) Quality control of EV-depleted medium was performed in our previous experiments both by dedicated flow cytometry and by TEM analysis (unpublished data, image reported below). Specifically, EVs-depleted medium obtained by overnight ultracentrifugation was used to treat THP1-derived macrophages (M0) for 24 hours. At the same time, to determine any serum contaminants, EVs depleted medium was cultured in absence of cells for 24 hours (EVs-DEP Medium). M0 cells conditioned cell culture media and EVs-DEP Medium were processed by UF/SEC. The obtained fractions were labeled with fluorophore conjugated lactadherin and analyzed by dedicated flow cytometry as reported in de Ron et al., 2019 (doi: 10.1080/20013078.2019.1643671). As shown in the figure (panel a), the fractions from the conditioned medium of M0 cells richest in lactoadherin+ particles were fractions 8, 9 and 10. Importantly, the same number of particles was not determined in the fractions obtained from EVs-DEP medium; this suggests a good depletion of serum EVs by overnight ultracentrifugation (panel a). Using refractive index (RI) to evaluate lipoproteins contamination, lipoproteins concentration in the fractions obtained from both M0 and EVs-DEP medium was drastically lower than EVs lactdherin+ particles (panel b).  To confirm these data, a quality check of the fractions (F8-F10) obtained from WT SHSY5Y conditioned medium and EV-depleted medium (without cells) was conducted by TEM analysis (panel c). As shown in the figure (panel c), EVs were only detected in the WT SHSY5Y fractions and not in those obtained from the EV-depleted medium.
  6. The protocol used is described and characterized in the reference 24 added to the manuscript.
  7. The term “massive” has been deleted.
  8. The word has been replaced as requested.
  9. We agreed with the referee and added the missing information in the text, as shown in the histograms, the total number of vesicles were: [figure 2: 59 for the WT CTR, 47 for the WT H2O2, 43 for the G93a CTR and 47 for the G93a H2O2]; [figure 3: 569 for the WT CTR, 566 for the WT H2O2, 258 for the G93a CTR and 470 for the G93a H2O2]; [figure 4 (Nu and Pn/CE): 243 and 293 for the WT CTR, 273 and 292 for the WT H2O2, 62 and 60 for the G93a CTR, 60 and 60 for the G93a H2O2]  Practically speaking the statistical relevance of the results that we obtained in our experiment was very far to be ‘artifactual’ since the acquisition of high-resolution AFM images is definitively not the right technique to measure a great number of vesicles. Practically speaking, to obtain a huge number of vesicles, each sample should be imaged for one month at least and this is almost a non-sense. We decided to acquire a sufficient number of images for each sample of isolated vesicles and stop the acquisition when we reached a well-defined distribution of vesicles (in general around 250-600 vesicles for each sample, that means roughly from 20 to 30 images for each sample). For the cells, instead, starting from our previous experimental experience we decided to image 10 randomly selected cells for each sample. It should be noted that with the AFM it is almost impossible to image a sufficient number of cells that is normally considered a ‘population’ from the biological point of view, but the few cells that are normally analyzed, are taken at very high lateral resolution (that is the great advantage of using the AFM technique) and this aspect grants us results that were statistically significant.
  10. Since there were no TEM images related to the cell body and protrusions of all experimental groups to support our description, we modified the TEM panel in Figure 1 and rearranged the text without referring to the different parts of the cell (e.g. cell body and protrusions).
  11. We have provided new EM images from a new EVs preparation (NEW Figure 2). 

    Although sizing conducted with TEM microscopy may slightly underestimate the diameter of isolated EVs because of sample preparation, several papers show its greater accuracy than using the NTA standard. In particular:

    - a work by E. Van der Pol et al., 2014 (DOI: 10.1111 / jth.12602) showed how, by measuring the size distribution of standard polystyrene nanobeads of different diameters, the relative error of the size and the coefficient of variation were the most variable when compared to other measurement techniques (i.e., TEM , RPS, conventional flow cytometry and dedicated flow cytometry). In addition, the same paper points out that NTA was the method with the lowest accuracy in determining the size due to the presence of large particles (from 200-400 nm), with an extensive broadening and an overestimation of particle concentration of large vesicles when compared with the other methods; 

    - in another review by Bağcı et al., 2022, the different limitations of the NTA and how the technique distorts the detection of larger particles is well described.

    Furthermore, in our work the intent is to compare two high resolution microscopy techniques (TEM and AFM) for the sizing of isolated vesicles starting from the same sample preparation. This approach is useful for a comparison based solely on the resolution of the microscopy method used. NTA is not useful for our purposes since the sample is measured in solution in its native form (therefore a different starting sample) and, as an optical method, a comparison with microscopy-based analysis is not useful.  

    We disagree with reviewer 2 in the use of markers (e.g., CD9, CD63, etc.) as a quantitative analysis of EVs, as the enrichment of a protein is not necessarily proportional to the amount of isolated vesicles. "A single vesicle from one group of cells could contain many more proteins than 10 EVs isolated from another group." From our side, we think that the use of markers is useful for a qualitative investigation to determine the purity of the isolated fractions. Indeed, in this work we have not used the EV markers for quantitative purposes, but only to determine the presence of EVs in the isolated fractions.

  12. The word has been replaced as requested.
  13. Using AFM analysis, budding vesicles are easily distinguished from other modifications of the cell membrane by their relatively regular morphology, and this discrimination is further strengthened if both z values and diameters of the buds are taken into account. We agree with the referee that the errors associated with the median value reported were missing. We added this information in the text and now it is more clear why we obtain statistically significant results. For example now the line 346 is: 297,4 +/- 4.7 nm versus 334 +/- 5.3 nm. The different areas were identified by using the morphology of the cell itself as measured with the very same AFM image, as it is commonly known in high-resolution microscopy the nuclear area of an intact cell adhered to a substrate is higher with respect to the surrounding cytoskeleton.

Reviewer 3 Report

Authors are presenting a new approach to the characterization of the extracellular vesicles using AFM. As the study îs relativelly simple, could be helpfull in development of new techniques of investigații.

Author Response

We thank the reviewer 3 for the comments.

Reviewer 4 Report

The paper titled “Wild-Type and SOD1-G93A SH-SY5Y under oxidative stress: EVs characterization and topographical distribution of budding vesicles” by Sbarigia et al.. used AFM to investigate the EV before and after the H2O2 treatment. The author has done a good work on the morphological characterization of EVs. This work has been done well. However some related issues need to be solved before the publication.

1. Some works have been done the AFM characterization of the EVs related. the authors should point out them well in the introduction part. The now presented introduction is not well.

2. I did not understand the reason of using the H2O2 as the control experiment. The authors need to tell the detail in the introduction part.

3. the Supplementary Materials with no caption. The authors need to give a file and using the caption. The present of a picture is bad.

4. the AFM can not only give about the morphological information but also the charge information known as the KPFM. In this manuscript, the authors do not need to do that. In the future, it is a good direction.

5. the afm only gives the information of a small part. The result is so reliable. The authors need interpret that in the introduction part.

Author Response

  1. We introduced in the introduction section a statement referring to some papers that have performed AFM to characterize EVs.
  2. As described in the discussion, we performed multiple comparisons to investigate not only the responsiveness of each cell type to H2O2 (WT CTRL vs. WT H2O2; SOD1-G93A CTRL vs. SOD1-G93A H2O2) but also to emphasize that the cells carrying the mutation in SOD1-G93A, both without treatment (WT CTRL vs. SOD1-G93A CTRL) and after exposure to H2O2 (WT H2O2 vs. SOD1-G93A H2O2), show a different behavior (in terms of vesicle budding localization and entity) with respect to the WT counterpart. The differences we found may suggest that the single presence of the mutation is sufficient to observe a different response by the cells.
  3. The caption was wrongly introduced in the “Supplementary Materials” section. Another file of the supplementary material with the caption was presented.
  4. We fully agree with reviewer 3's suggestion. Indeed, our future investigations will be directed towards a correlation between the topographic analysis of surface budding, mechanical properties of cells and surface potential.
  5. In the introduction has been included a statement referring to the need to further investigate the biological significance of the changes we have detected with AFM. The point of our work is the validation of an EM/AFM approach to study variations in the EVs trafficking and represents a start point to detect anomalies in the cell physiology.

Round 2

Reviewer 2 Report

I would like to thank the authors for their efforts to answer the issues I had with the first version. While I am still unconvinced that their AFM approach truly measures EV release (because the authors are ignoring other cell-surface processes), I found the manuscript improved and their arguments convincing.

One remaining (important) issue: the authors provide new EM images of a new EV preparations (new Fig 2A). However the EV size quantifications and Western blots were performed with the previous samples that were of poor quality. The authors should provide new quantifications and western blots with the new samples.

Author Response

New western blots and new size distributions were performed and included in the manuscript as requested. 

Reviewer 4 Report

can be accepted

Author Response

We thank the reviewer for the comments.